# Screening for NAFLD—Current Knowledge and Challenges

**DOI:** 10.3390/metabo13040536

**Published:** 2023-04-09

**Authors:** Roberta Forlano, Giordano Sigon, Benjamin H. Mullish, Michael Yee, Pinelopi Manousou

**Affiliations:** Liver Unit, Division of Digestive Diseases, Department of Metabolism, Digestion and Reproduction, Faculty of Medicine, Imperial College London, London W21NY, UK

**Keywords:** non-alcoholic fatty liver disease, screening, primary care, liver fibrosis

## Abstract

Non-alcoholic fatty liver disease (NAFLD) is the most common cause of abnormal liver function tests worldwide, with an estimated prevalence ranging between 19–46% in the general population. Of note, NAFLD is also expected to become a leading cause of end-stage liver disease in the next decades. Given the high prevalence and severity of NAFLD, especially in high-risk populations (i.e., patients with type-2 diabetes mellitus and/or obesity), there is a major interest in early detection of the disease in primary care. Nevertheless, substantial uncertainties still surround the development of a screening policy for NAFLD, such as limitations in currently used non-invasive markers of fibrosis, cost-effectiveness and the absence of a licensed treatment. In this review, we summarise current knowledge and try to identify the limitations surrounding the screening policy for NAFLD in primary care.

## 1. Definition and Epidemiology of NAFLD

Non-alcoholic fatty liver disease (NAFLD) is the most common cause of abnormal liver function tests worldwide, with a global estimated prevalence of 30% [1]. NAFLD encompasses a spectrum of pathological disorders characterised by macro-vesicular fat accumulation (steatosis, non-alcoholic fatty liver, NAFL) with or without hepatocellular injury and/or inflammation (non-alcoholic steato-hepatitis, NASH) and a variable degree of fibrosis up to cirrhosis [2,3].

Overall, NAFLD prevalence is particularly high in those with metabolic syndrome, i.e., a combination of central obesity, insulin resistance, type 2 diabetes mellitus (T2DM), hypertension and dyslipidaemia [4]. According to tertiary care studies, more than 50% of the patients with T2DM have NAFLD [4]. Similarly, the prevalence of NAFLD is as high as 45% among those with increased body mass index (BMI > 30 kg/m^2^) and up to 90% among those undergoing bariatric surgery (BMI > 35 kg/m^2^) [5]. Mirroring the epidemic of metabolic syndrome, the prevalence of NAFLD is constantly increasing in the general population, increasing from 33% in 2005 to 59.1% in 2010, and in similar fashion, the prevalence of NASH increasing from 15% to 25% [6]. The total NAFLD population in 2015 was estimated at 83.1 million cases, which is projected to increase by 21% to 100.9 million cases by 2030 [7]. Furthermore, NAFLD/NASH has become the fastest growing indication for liver transplantation in the USA [8]. According to data from the European Liver Transplant Registry, the proportion of liver transplants performed for NASH increased from 1.2% in 2002 to 8.4% in 2016 [9].

## 2. Pathogenesis of NAFLD

Far from the old concept of being a dichotomic disease, NAFLD is now considered to be a dynamic disease with a wide spectrum of disease activity within different stages of simple steatosis or NASH [10]. Insulin resistance plays a crucial role in the development and progression of liver disease, as this stimulates de novo lipogenesis and is associated with impaired lipolysis, resulting in an increased flux of fatty acid to the liver [11]. Of note, hepatic triglyceride storage is not harmful per se. Nevertheless, when the hepatic capacity of using, storing, and exporting free fatty acids becomes saturated, lipotoxicity may occur within the liver. Lipotoxicity is thought to be the crucial driver for the development and progression of hepatocellular injury, inflammation, hepatic stellate cell activation and extracellular matrix deposition, leading to fibrosis progression [12]. Overall, a status of insulin resistance also drives a dysfunctional adipose tissue, which produces metabolically active cytokines and initiates an inflammatory cascade [13] (Figure 1).

The development and progression of liver disease in NAFLD is now being explained with the multi-hit hypothesis, where a plethora of modifiable (dietary and environmental) and non-modifiable (genetic) factors contribute to the disease along with the worsening of insulin resistance (Figure 1). Among the modifiable factors, dietary elements, both in terms of overall calorie intake and specific dietary patterns, may contribute to the development of NAFLD [14,15]. Specifically, a high-fat diet and increased fructose and red meat intake have been associated with worsening hepatic steatosis and the induction of a pro-inflammatory status [16,17,18]. More recently, there has been a large body of work showing that gut microbiome plays an essential role in disease activity in patients with NAFLD [19]. The interactions between the liver and the gut—the so called “gut–liver axis”—result from a complex interplay between the gut and the immune system, which ranges from immune tolerance to immune activation. Changes in gut microbiome composition [20], gut permeability [21], and the translocation of pro-inflammatory bacterial by-products [22] are now included among the factors involved in the progression of liver disease in this population [23]. Among the non-modifiable factors, genetic factors also represent an important contribution to NAFLD progression. Few genes have been identified as conferring different levels of susceptibility to fat accumulation, hepatic inflammation and lipotoxicity, such as patatin-like phospholipase domain containing protein 3 (PNPLA3) [24,25], transmembrane 6 superfamily member 2 (TM6SF2) [25] and membrane-bound O-acyltransferase domain containing 7 (MBOAT7) [26], with many more in the pipeline to be identified.

## 3. Natural History

Clinical data from paired liver biopsies and an analysis of the biopsies from placebo arms of clinical trials have demonstrated that up to 25% of patients with NAFL, a condition which was previously considered “benign”, may also progress to advanced fibrosis [27]. Among those with NASH, up to 35% present fibrosis progression, while 40% remain relatively stable over time [28]. Overall, NAFLD is considered a slow progressive disease, with a one-stage fibrosis progression over 14 years for those with fibrosis stage 1 and over 7 years in patients with advanced fibrosis [29]. However, those with NASH seem to progress more rapidly than those with NAFL [30], with up to 20% of patients with NASH and fibrosis stage 3 progressing to cirrhosis [31,32]. Baseline inflammation status and uncontrolled metabolic risk factors have been suggested as the main contributors to disease progression in these patients [29]. Overall, the yearly cumulative incidence of NASH-related hepato-carcinoma (HCC) is low (1.5–2%) compared to 4% of HCC from chronic viral hepatitis [7,32]. Moreover, recent evidence also suggests that pre-cirrhotic NAFLD may confer an increased risk for HCC, independent of cirrhosis [7] (Figure 2). Interestingly, NAFLD-associated HCC in non-cirrhotics accounts for up to 25–46% of all NAFLD-associated HCC cases, and the incidence is estimated at approximately 0.1 to 1.3 per 1000 patient-years [33,34,35].

The main cause of death among NAFLD patients is cardiovascular disease (CVD) (33% of deaths), followed by extra-hepatic cancer (19% of deaths) and liver-related complications (19% of deaths) [36]. In terms of absolute risk, patients with NAFLD have been shown to have significantly increased risk of all-cause mortality (hazard ratio 2.2, 95%CI 1.2–44), mainly driven by malignancy [36]. However, CVD is still highly prevalent and represents an important cause of mortality and morbidity in these patients [37,38]. Previously published studies have explored clinical, biochemical, and histological variables that could predict mortality in patients with NAFLD, concluding that age and T2DM are strong predictors for adverse events [39]. However, it is now established that fibrosis stage represents the main prognostic factor in this population [40]. Nevertheless, obesity seems to be the main mediator of the increased risk of malignancies in this population, especially with regards to colon and breast cancer [41]. However, there is evidence suggesting that NAFLD may be associated with malignancies also in those with a lean phenotype [42].

**Figure 2 metabolites-13-00536-f002:**
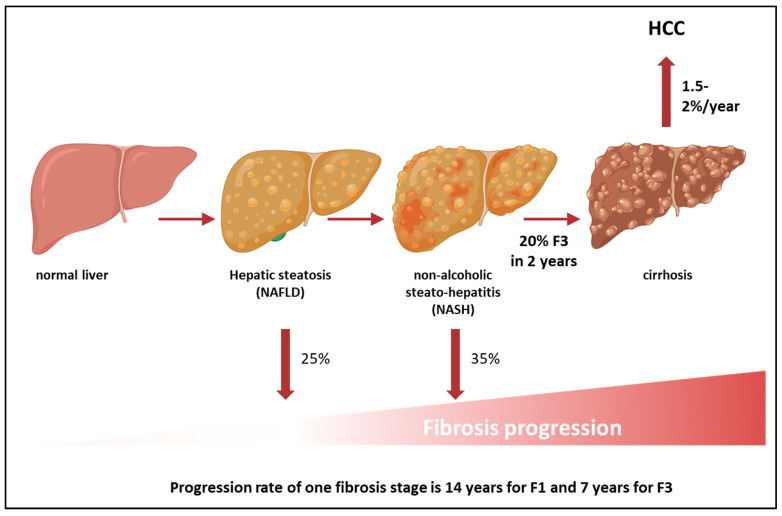
Natural history of NAFLD. NAFLD affects almost 25% of the population worldwide. Up to 25% of patients with simple steatosis and up to 35% of patients with NASH may develop cirrhosis [32,43]. Overall, progression rate for one fibrosis stage is 14 years for F1 and 7 years for F3. However, a subgroup (20%) of patients with NASH and F3 may be fast progressors and develop cirrhosis in 2 years. The overall yearly incidence of HCC in those with NASH cirrhosis is 1.5–2%. The incidence of HCC among patients with non-cirrhotic NAFLD is lower, approximately 0.1 to 1.3 per 1000 patient-years [35]. Abbreviations: NAFLD: non-alcoholic fatty liver disease, NASH: non-alcoholic steatohepatitis, F3: fibrosis stage 3, F1: fibrosis stage 1.

## 4. Diagnosis and Staging of NAFLD

Most patients presenting with NAFLD are mainly asymptomatic. Hepatomegaly (liver enlarged in size) is the most common clinical finding on physical examination. As patients progress to advanced liver disease, signs and symptoms related to portal hypertension may become more evident. A diagnosis of NAFLD should be suspected in all patients with at least one component of the metabolic syndrome presenting with evidence of hepatic steatosis on imaging. Notably, NAFLD is still a diagnosis of exclusion of other common causes of liver disease, especially steatogenic medications and chronic alcohol consumption [4,44].

Histology assessed by an expert pathologist remains the gold standard for diagnosing NASH and staging liver disease in NAFLD [4,11,44]. Currently, the interpretation of liver biopsy relies on the use of semi-quantitative scores, such as the NASH clinical research network (NASH CRN) scoring system. When using such scores, disease activity is defined based on steatosis, hepatocyte ballooning and lobular inflammation, while staging is based on the assessment of fibrosis [2]. Nevertheless, obtaining a liver biopsy is expensive, invasive and associated with potential complications (i.e., bleeding, pain). More importantly, considering the high prevalence of the disease, histology cannot be considered in all patients but should be limited to selected sub-groups, such as those at high risk for progressive disease, and/or in the setting of clinical trials [10,45].

## 5. Non-Invasive Assessment of NAFLD

Given the drawbacks of performing a liver biopsy, there has been an explosive development and use of non-invasive tests (NITs), with significant (fibrosis stage F ≥ 2) and advanced (fibrosis stage F ≥ 3) fibrosis being the main endpoints [44,46]. A large number of blood and imaging-based NITs are now available to use in clinical practice.

Overall, serum NITs are mainly clustered into two groups: direct (or class 1) and indirect (or class 2) biomarkers (Table 1). The direct NITs measure substances that are directly associated with the process of fibrogenesis in the hepatic stellate cells. Conversely, indirect NITs consist of a combination of biochemical tests, such as LFTs, platelet (PLT) count and/or albumin, along with patient demographics, such as age, BMI and/or the presence of T2DM. There are also patented NITs, which come from a combination of class 1 and class 2 biomarkers. As histology is the gold standard to assess liver disease severity, NITs have been developed using liver biopsy as a gold standard. While direct biomarkers usually have a single cut-off with high specificity, indirect biomarkers usually provide two: a low cut-off with high sensitivity (to rule out the disease) and a high cut-off with high specificity (to rule in the disease). The use of each cut-off is mainly dictated by the clinical setting and/or the disease prevalence. If these cut-offs are combined, the number of false positive and false negative tests are usually reduced. However, when applying such a system, a significant sub-group of patients would inevitably fall in the indeterminate-risk group and, therefore, will require further investigations similar to those in the high-risk category. Among others, the Fibrosis-4 score (FIB-4) [47], NAFLD fibrosis score [48] and enhanced liver fibrosis (ELF) score [49] are the most frequently used NIT measures in clinical practice. Such non-invasive markers have now been embedded in the daily assessment of patients at high risk for NAFLD and suggested for screening for fibrosis in this population.

Imaging has also made huge advancements in the field of non-invasive assessment of liver disease in NAFLD patients. Among others, elastographic techniques have revolutionized the management of these patients, combining high sensitivity and specificity for predicting liver fibrosis. Transient elastography (TE) (Fibroscan, Echosens, Paris), acoustic radiation force imaging (ARFI), shear wave elastography and magnetic resonance elastography (MRE) are by far the most popular in the field [50,51]. Specifically, TE employs vibrations of mild amplitude and low frequency, which propagate within the liver. The subsequent pulse-echo acquisitions are reflective of the elastic properties (i.e., stiffness) of the liver, which are expressed in kilopascals (kPa). From a patient’s perspective, TE is painless and rapid (<5 min) and thus highly acceptable. From a clinical perspective, TE provides high accuracy and reproducibility for detecting advanced liver fibrosis [51]. As such, TE has now become the NIT of choice in most liver clinics around the world.

The choice of NIT used in clinical practice is based on different factors, such as the availability and cost of the test/technique as well as the “context of use”. For instance, class 2 biomarkers, which require inexpensive and widely available parameters, can be easily used to predict liver fibrosis in large populations (i.e., primary care). Conversely, sophisticated, time-consuming and expensive techniques like MRE are applied in selected groups of patients and for research purposes (i.e., tertiary care) [52,53].

## 6. Screening for NAFLD in Primary Care: Current Recommendations

Given the high prevalence and severity of NAFLD in those with metabolic syndrome and type-2 diabetes, there is an expected large burden of undiagnosed NAFLD with advanced fibrosis in the community, and—as such—a major interest in early detection of the disease in primary care [44]. Furthermore, as there is no licensed treatment for the disease, early detection of fibrosis is of the utmost importance in this population. For instance, in the United States, it has been estimated that up to 9 million diabetic patients have NASH, while 4 million are at risk for advanced fibrosis [54]. Similar evidence has been made available for those suffering from obesity [55].

The latest European guidelines recommend screening NAFLD in high-risk populations (i.e., patients with metabolic syndrome) following a two-tier system (Figure 3). Specifically, it is recommended that patients should be stratified using FIB-4 and/or ELF in primary care, followed in sequence by TE in a specialist setting [44]. The most recent guidelines from the America Association for the study of the liver diseases (AASLD) suggest yearly testing with FIB-4 for diabetics and those with at least two components of metabolic syndrome, whilst not recommending screening for NAFLD in the general population [46]. Interestingly, the latest UK NICE guidelines [56] recommend screening for NAFLD subjects with T2DM and metabolic syndrome, including LFTs and/or ultrasound. Similar recommendations are made in the guidelines from the Asian Pacific Association [57] and from the Latin American Association [58] for the study of the liver. However, LFTs assessment is not sufficient alone for screening NAFLD, since it is well established that NASH and significant fibrosis can occur in patients with normal range LFTs [59,60]. Furthermore, ultrasound has low reproducibility and was not designed to stage disease severity [44].

Unfortunately, there is a substantial lack of awareness among policy makers outside the hepatology community. For instance, current diabetes and obesity management guidelines do not advise for NAFLD screening in the respective populations [61]. Nevertheless, the American Association of Clinical Endocrinology has now published a clinical practice guideline on the management of NAFLD in primary care and endocrinology settings, opening the door to future joint position papers across different specialties [62]. Such guidelines highlight that patients at high risk of NAFLD do not require an abdominal ultrasound to diagnose hepatic steatosis; it is recommended to move directly to risk stratification. Finally, European guidelines on obesity care in patients with chronic GI conditions, despite recognising that obese patients should be screened for NAFLD, do not advise fibrosis risk stratification [63].

## 7. Screening for NAFLD in Primary Care: Limitations

Although screening for NAFLD in high-risk populations has been supported by EASL and AASLD guidelines, a consensus on the cost-effectiveness of screening has not yet been reached. Corey and colleagues performed a simulation to compare quality-adjusted life years (QALYs) between screening with liver biopsy and non-screening, including pioglitazone as therapeutical option. The authors reported that NASH screening could have been cost-effective if superior treatment had been made available at the time of the model [64]. A recent paper suggested that for a pharmacological intervention to be cost-effective in the NAFLD fibrosis population, the annual drug cost should not exceed $12,000 per patient [65]. Several studies tried to analyse the cost-effectiveness of screening by factoring in the effect of early detection in slowing disease progression rather than the therapeutical effect of a new pharmacological agent. A recent cost-utility analysis also demonstrated that screening patients with T2DM with US and LFTs, followed by non-invasive tests, was more cost-effective than not screening [66]. Interestingly, a UK-based study comparing risk stratification using TE vs. standard of care proved to be cost-effective in the general population [67]. Similarly, in a study conducted in the US health system, screening for NAFLD cirrhosis with FIB-4, followed by TE and liver biopsy, was more effective than FIB-4 followed by MRE [68]. Another study demonstrated that FIB-4 followed by share wave elastography was the most effective and least costly strategy in the community [69]. Nevertheless, the lack of licensed pharmacological treatment still represents an important limitation to establishing the cost-effectiveness of screening in this population. Furthermore, a recent metanalysis also pointed out how the currently used health economic models are associated with limitations, primarily driven by a lack of NASH-specific data [70].

Primary care clinicians play an essential role in identifying patients with NAFLD who are at risk of significant liver disease [44]. In this sense, the Lancet commission on liver disease identified the need for streamlined diagnostic pathways for screening people with NAFLD as a priority area to defeat liver disease [71]. However, an important limitation to screening pathways of NAFLD is an overall low awareness among primary care clinicians, possibly as the result of gaps in knowledge as well as lack of awareness of relevant practice guidelines. In a survey study, over 40% of general practitioners (GPs) were not familiar with clinical published guidelines for NAFLD management [72]. Moreover, GPs were more likely to screen low-risk patients while neglecting patients at high risk for liver fibrosis. Again, this phenomenon has been attributed to the misconception that LFTs may reflect disease severity. On a similar note, a UK-based qualitative study demonstrated that the diagnosis and management of NAFLD is perceived as a great challenge by GPs [73,74]. Overall, less than 3% of patients with elevated FIB-4 are currently referred to the specialist setting for further investigations [75], with GPs not perceiving NAFLD as a priority in their clinical activities [76].

Low standardisation of the screening protocols also represents an important limitation to screening NAFLD in the community. Several studies have highlighted a significant gap between guidelines and real-life clinical approaches, not only across different continents [77] but also within Europe [78]. Such inconsistency translates into a lack of clarity for primary care physicians. The use of an automated calculator for NITs as well as easier access to second-line non-invasive tests have been identified as possible strategies to overcome current barriers to screening [76]. Clear guidance on the groups to be screened and the patients to be referred for further tests also appears to be lacking. A previous study carried out in Scotland demonstrated how an algorithm for analysis of abnormal LFTs was found to correctly (in 91.3% of cases) stratify patients for referral to specialist investigation [79]. A similar approach could be potentially useful for identifying those at higher risk of fibrosis from NAFLD. Furthermore, NAFLD screening could be embedded in the routine clinical management of high-risk populations in primary care, such as those with type-2 diabetes [76].

Along with developing cost-effective screening for NAFLD in primary care, future work should also focus on education regarding high-risk stratification, providing easy-to-use tools and building awareness among primary care physicians.

## 8. Screening for NAFLD in Primary Care: Beware of the Spectrum Effect

Ideally, screening tests should be derived from a cohort that mirrors the target population so that spectrum biases can be minimised [80]. From an epidemiological perspective, the spectrum effect describes the variation in the diagnostic performance of predictive tests when applied to populations with different disease prevalence. Due to the spectrum effect, NITs will have lower sensitivity and higher specificity in populations with lower disease prevalence. On the other hand, in secondary/tertiary care settings (higher disease prevalence), the positive predictive value will be higher, as the probability of observing true positive cases is higher a priori. Among others, NITs based on blood tests and/or on a combination of clinical features seem to be particularly affected by the spectrum bias.

Notably, most of the NITs used for NAFLD screening were historically developed and validated in secondary or tertiary care settings. Their performance in primary care is largely unknown. Specifically, FIB-4 was developed from a cohort of patients with biopsy-proven chronic hepatitis C [81], while the NAFLD fibrosis score and the ELF were developed in a cohort of patients with biopsy-proven NAFLD [48,82]. Of note, in the pre-elastography era, the average biopsy patient was selected based on clinical parameters, mainly on LFTs. It is therefore not surprising that these cohorts were characterised by older age and elevated LFTs, despite these not necessarily being a good marker of the disease severity [60]. It is therefore expected that FIB-4, NAFLD fibrosis and ELF score captures the phenotype of patients referred to specialist clinics.

The EASL algorithm for NAFLD screening has recently been validated into a tertiary care cohort, despite primary care being the main target for the pathway [83]. Interestingly, a recent real-life study showed that up to two-thirds of the new referrals to hepatology clinics are discharged after their first assessment, suggesting that current risk stratification needs optimisation [84]. Furthermore, there is emerging evidence suggesting that FIB-4 accuracy is much lower in the community [85], especially when used to assess young patients with normal liver function tests [85,86]. Moreover, a recent metanalysis also demonstrated that ELF performance is not consistent across studies [87], suggesting that dedicated cut-offs may be needed for different populations. On this note, the most recent AASLD guidelines have highlighted the lack of evidence to support the use of some NITs in primary care, raising concerns about underestimating liver disease, especially among diabetics [46].

Overall, there is increasing evidence suggesting that FIB-4, ELF and NAFLD fibrosis scores may be affected by the spectrum effect. Offering TE to high-risk patients in primary care could represent a way forward, as it is cost-effective and is not affected by spectrum biases [86]. Future work should focus on assessing the performance of NITs in true primary care cohorts and on the optimization of current referral management strategies.

## 9. Novel Approaches to Diagnosing and Staging NAFLD

In the last decade, several new technologies have been developed with the aim to improve precision medicine in the field of metabolic diseases. Among others, metabolomics has been considered as a potential new approach for diagnosing and staging NAFLD. In a recent study, disease activity, assessed by hepatocellular ballooning and inflammation as per the NASH CRN scoring system, correlated with increased branched chain amino acids and aromatic amino acids. In the same study, a combination of glutamate, serine and glycine could predict fibrosis severity [88]. In another study, NASH could be diagnosed accurately by a combination of glycocholic acid, taurocholic acid, phenylalanine and branched chain amino acids [89].

Similarly, lipidomics has also been explored for the non-invasive assessment of NAFLD [90]. Previous studies demonstrated that a combination of circulating lipids may be able to accurately identify those with NASH [90,91]. Similarly, another algorithm combining genetic variants and serum lipids correlated with hepatic fat fraction, as measured by MRI-PDFF [92]. Finally, among those with biopsy-proven NASH, phosphatidylcholine levels were strongly associated with disease activity, as assessed by the NASH CRN scoring system [93].

Future studies will be required to validate the findings from lipidomic and metabolomic studies on a large scale and against histology. Despite the high cost being a potential limitation to applying these techniques on a large scale, obtaining an accurate, non-invasive diagnosis of NASH may fill a critical gap in clinical practice.

## 10. Conclusions

Non-alcoholic fatty liver disease poses a significant challenge to the Hepatology community due to increasing burden of the disease. As many promising drugs are in the pipeline, identifying those with advanced fibrosis or at risk of developing advanced fibrosis in the community will become a clinical priority in the near future. Engaging with primary care is crucial, as GPs are at the forefront of identifying patients with NAFLD in need for further evaluation. There is a need for a simple, pragmatic referral/management pathway that performs well in primary care and that could be easily implemented. Future work should focus on the optimisation of current algorithms for NAFLD screening.

## Figures and Tables

**Figure 1 metabolites-13-00536-f001:**
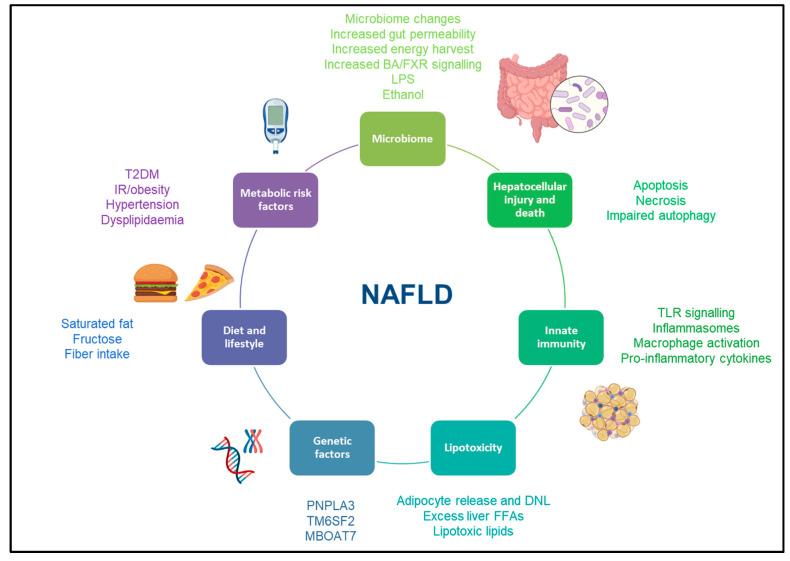
Risk factors for development and progression of NAFLD. This figure illustrates the most common risk factors associated with the development and progression of NAFLD. Abbreviations: NAFLD: non-alcoholic fatty liver disease, BA: bile acid, FXR: farnesoid X receptor, LPS: lipopolysaccharide, TLR: toll-like receptor, DNL: de novo lipogenesis, T2DM: type 2 diabetes mellitus, IR: insulin resistance, PNPLA3: patatin-like phospholipase domain containing protein 3, TM6SF2: Transmembrane 6 superfamily member 2, MBOAT7: membrane-bound O-acyltransferase domain containing 7.

**Figure 3 metabolites-13-00536-f003:**
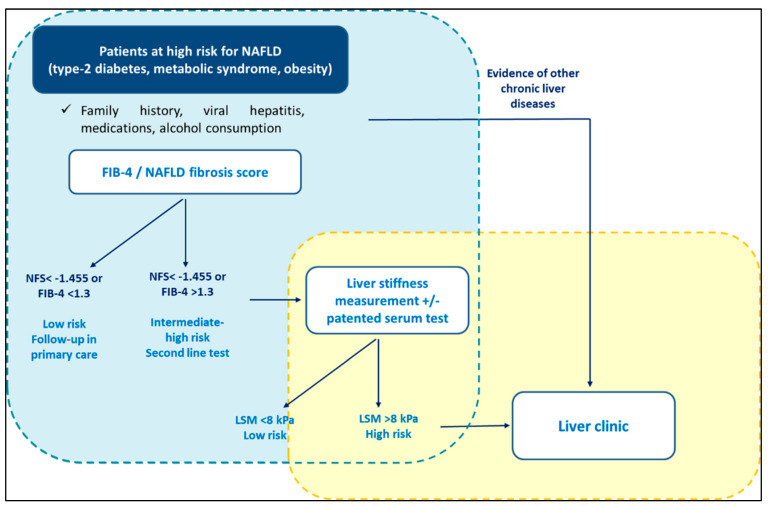
Screening for NAFLD in primary care. International guidelines recommend screening for NAFLD in high-risk populations (i.e., patients with metabolic syndrome) following a two-tier system.

**Table 1 metabolites-13-00536-t001:** Overview of the main non-invasive tests for NAFLD based on blood tests and clinical parameters.

*Direct serum markers*	Hyaluronate Laminin Chitinase-like protein 40 (YKL-40) Procollagen type I carboxy-terminal peptide (PICP) Procollagen type III amino-terminal peptide (PIIINP) Metalloproteinases (MMP)-1 and MMP-2 Tissue inhibitors of the metalloproteinases (TIMPs) Transforming growth factor β1 (TGF-β1) Microfibril-associated glycoprotein 4 (MFAP-4)
*Indirect serum markers*	AST/ALT ratio Prothrombin time, γ-glutamyl transferase and apolipoprotein A1 (PGA) APRI Forns index Fibrosis-4 (FIB-4) Lok index Fibrosis probability index (FPI) NAFLD fibrosis score BARD Gamma-glutamyl transferase (GGT) to platelet (PLT) ratio
*Patented serum markers*	Fibrotest Fibroindex Hepascore Fibrospect Enhanced liver fibrosis (ELF) test Fibrometers

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
