# Peer review of "Screening for NAFLD—Current Knowledge and Challenges"

_metabolites, 2023, doi:10.3390/metabo13040536_

Round 1
Reviewer 1 Report
The bibliography used in the work is adequate and current. In addition, the title is in agreement with the text and the conclusions emerge from the review.
Author Response
Reviewer 1.
The bibliography used in the work is adequate and current. In addition, the title is in agreement with the text and the conclusions emerge from the review.
Thank you very much for the kind comments on our work.
Reviewer 2 Report
A valuable review. Please consider the following comments:
L. 2 explain abbreviation.
L.2 and 20, is the prevalence given in l.2 among all people or only among those with abnormal LFTs?
L.7 explain "LFTs" (write out).
L.21 "vesicular"
L.24 "due to", not through
L.30 "5" correct reference?
L.33 How can prevalence of NASH be higher than that of NAFLD (l.20)?
L.40 "in", not into.
L.77 "and placebo arms", not understandable.
L.92, "the is"-meaning?
L.114 explain the abbreviation HCC
L.145-148 and Table 1; it is confusing that a direct NIT measure can be indirectly associated with fibrogenesis, and that also indirect NITs exist?
L.158 "as per"?
L.159-160 the FIB-4 and ELF scores are said to be important on p.6. Accordingly, they should be explained in more detail.
L.183 tests of NAFLD?
Table1 abbreviations (e.g. PGA) should be explained. What is meant by "Patented"?
L.191 ref for European guidelines?, 43?
L.192 "a 2-tiers system"-meaning?
L.196 "components"
L203 full stop, not comma
L.226"is that per patient?"- please answer! "forward"
L230 "more effective", in terms of?
L.265 "triage", I do not know the word, refer?
Author Response
Reviewer 2.
A valuable review. Please consider the following comments:
- 2 explain abbreviation.
We have addressed this comment.
L.2 and 20, is the prevalence given in l.2 among all people or only among those with abnormal LFTs?
We have addressed this comment.
L.7 explain "LFTs" (write out).
We have addressed this comment.
L.21 "vesicular"
We have addressed this comment.
L.24 "due to", not through
We have addressed this comment.
L.30 "5" correct reference?
We have changed the reference with doi: 10.1016/j.diabet.2022.101363
L.33 How can prevalence of NASH be higher than that of NAFLD (l.20)?
We have addressed this comment.
L.40 "in", not into.
We have addressed this comment.
L.77 "and placebo arms", not understandable.
We have addressed this comment.
L.92, "the is"-meaning?
We have addressed this comment.
L.114 explain the abbreviation HCC
We have addressed this comment.
L.145-148 and Table 1; it is confusing that a direct NIT measure can be indirectly associated with fibrogenesis, and that also indirect NITs exist?
We have addressed this comment.
L.158 "as per"?
We have addressed this comment.
L.159-160 the FIB-4 and ELF scores are said to be important on p.6. Accordingly, they should be explained in more detail.
We have addressed this comment.
L.183 tests of NAFLD?
We have addressed this comment.
Table1 abbreviations (e.g. PGA) should be explained. What is meant by "Patented"?
We have addressed this comment and explained all the acronyms included in the table
L.191 ref for European guidelines?, 43?
Yes, we confirm that the paper mentioned is the latest EASL guidelines on non-invasive markers of fibrosis
L.192 "a 2-tiers system"-meaning?
We have addressed this comment and made the concept clearer.
L.196 "components"
We have addressed this comment.
L203 full stop, not comma
We have addressed this comment.
L.226"is that per patient?"- please answer! "forward"
We have addressed this comment.
L230 "more effective", in terms of?
We have addressed this comment.
L.265 "triage", I do not know the word, refer?
We have addressed this comment.
Reviewer 3 Report
The review paper by Forlano et al. focuses on the screening of NAFLD. The paper is nicely written and easy to follow. Below are my comments and recommendations to further improve this review paper.
Formal comments/recommendations:
p. 1, l. 22: “Non-alcoholic fatty liver” correct to “non-alcoholic fatty liver”
p. 1, l. 23: “Non-Alcoholic Steato-Hepatitis” correct to “non-alcoholic steatohepatitis”
p. 1, l. 29: “Body mass index” correct to “body mass index”
p. 6, l. 194: “America Association of the study of the liver (AASLD)” correct to “American Association for the Study of Liver Diseases (AASLD)”
p. 6, l. 208: “American association of clinical endocrinology” correct to “American Association of Clinical Endocrinology”
Figure 1: Text in boxes (e.g., Microbiome); please, increase the font size. It is not easy to read this text.
p. 1, l. 23 & Figure 1: steato-hepatitis vs. steatohepatitis
Further comments/recommendations:
p. 4, l. 120: Explain “hepatomegaly” in the text (i.e., liver enlargement beyond its normal size).
p. 7, l. 226: “$12,000 (is that per patient?)” I presume this is the lifetime per-patient cost (based on the ref. [62]. I suggest modifying this sentence; “(is that per patient?)” looks like some internal note.
I recommend including a new chapter regarding changes in metabolites or complex lipids NAFLD using metabolomics and lipidomics approaches and possible implications of biomarkers (since your paper is also about challenges). Here are a few examples:
• Nonalcoholic fatty liver disease stratification by liver lipidomics (DOI: 10.1016/j.jlr.2021.100104)
• Lipidomics in non-alcoholic fatty liver disease (DOI: 10.4254/wjh.v12.i8.436)
• Metabolomics and lipidomics in NAFLD: biomarkers and non-invasive diagnostic tests (DOI: 10.1038/s41575-021-00502-9)
Specifically, in the paper Vvedenskaya et al. J Lipid Res 2021;62:100104 (DOI: 10.1016/j.jlr.2021.100104), there is a nice figure (Fig. 1. Lipid class composition of liver biopsies in the four main groups of patients) which would nicely fit this review paper so the readers will have the idea which lipid subclasses are not altered during NAFLD (e.g., LPC, LPG, LPI) and which ones increase (e.g., TG, DG).
Commenting on current NAFLD treatment in your paper would not be a bad idea. Some references are provided below:
• Current treatment of non-alcoholic fatty liver disease (doi: 10.1111/joim.13531)
• Emerging therapeutic approaches for the treatment of NAFLD and type 2 diabetes mellitus (DOI: 10.1038/s41574-021-00507-z)
• Is Vitamin E or Ursodeoxycholic Acid a Valid Treatment Option for Nonalcoholic Fatty Liver Disease in 2016? (doi: 10.4103/1319-3767.182462)
• Ursodeoxycholic acid alleviates nonalcoholic fatty liver disease by inhibiting apoptosis and improving autophagy via activating AMPK (10.1016/j.bbrc.2020.05.128)
Author Response
Reviewer 3.
The review paper by Forlano et al. focuses on the screening of NAFLD. The paper is nicely written and easy to follow. Below are my comments and recommendations to further improve this review paper.
Formal comments/recommendations:
- 1, l. 22: “Non-alcoholic fatty liver” correct to “non-alcoholic fatty liver”
We have addressed this comment.
- 1, l. 23: “Non-Alcoholic Steato-Hepatitis” correct to “non-alcoholic steatohepatitis”
We have addressed this comment.
- 1, l. 29: “Body mass index” correct to “body mass index”
We have addressed this comment.
- 6, l. 194: “America Association of the study of the liver (AASLD)” correct to “American Association for the Study of Liver Diseases (AASLD)”
We have addressed this comment.
- 6, l. 208: “American association of clinical endocrinology” correct to “American Association of Clinical Endocrinology”
We have addressed this comment.
Figure 1: Text in boxes (e.g., Microbiome); please, increase the font size. It is not easy to read this text.
We have addressed this comment.
|
|
|
|
|
|
|
|
Figure has been changed.
- 1, l. 23 & Figure 1: steato-hepatitis vs. steatohepatitis
We have addressed this comment.
|
|
|
|
|
|
|
|
Figure has been changed.
Further comments/recommendations:
- 4, l. 120: Explain “hepatomegaly” in the text (i.e., liver enlargement beyond its normal size).
We have addressed this comment.
|
|
|
|
|
|
|
|
Figure has been changed.
- 7, l. 226: “$12,000 (is that per patient?)” I presume this is the lifetime per-patient cost (based on the ref. [62]. I suggest modifying this sentence; “(is that per patient?)” looks like some internal note.
We have addressed this comment.
I recommend including a new chapter regarding changes in metabolites or complex lipids NAFLD using metabolomics and lipidomics approaches and possible implications of biomarkers (since your paper is also about challenges). Here are a few examples:
- Nonalcoholic fatty liver disease stratification by liver lipidomics (DOI: 10.1016/j.jlr.2021.100104)
- Lipidomics in non-alcoholic fatty liver disease (DOI: 10.4254/wjh.v12.i8.436)
- Metabolomics and lipidomics in NAFLD: biomarkers and non-invasive diagnostic tests (DOI: 10.1038/s41575-021-00502-9)
Specifically, in the paper Vvedenskaya et al. J Lipid Res 2021;62:100104 (DOI: 10.1016/j.jlr.2021.100104), there is a nice figure (Fig. 1. Lipid class composition of liver biopsies in the four main groups of patients) which would nicely fit this review paper so the readers will have the idea which lipid subclasses are not altered during NAFLD (e.g., LPC, LPG, LPI) and which ones increase (e.g., TG, DG).
Dear reviewer, thank you for this comment. We have implemented the suggestions adding a dedicated paragraph n.9. Novel approaches to diagnosing and staging NAFLD (page 11).
Commenting on current NAFLD treatment in your paper would not be a bad idea. Some references are provided below:
- Current treatment of non-alcoholic fatty liver disease (doi: 10.1111/joim.13531)
- Emerging therapeutic approaches for the treatment of NAFLD and type 2 diabetes mellitus (DOI: 10.1038/s41574-021-00507-z)
- Is Vitamin E or Ursodeoxycholic Acid a Valid Treatment Option for Nonalcoholic Fatty Liver Disease in 2016? (doi: 10.4103/1319-3767.182462)
- Ursodeoxycholic acid alleviates nonalcoholic fatty liver disease by inhibiting apoptosis and improving autophagy via activating AMPK (10.1016/j.bbrc.2020.05.128)
Dear reviewer, thank you for suggestions. We mentioned the crucial importance of early detection of NAFLD especially as pharmacological treatment is lacking (page 8,line 203-208). However, the authors feel that preparing a separate chapter on all the drugs in the pipeline goes beyond the scope of the review.
Reviewer 4 Report
In this literature review entitled “Screening for NAFLD – current knowledge and challenges” the authors provide a current update on NAFLD screening algorithms in the primary care setting and present as well as future challenges. The manuscript is well written and I have only minor comments or recommendations:
- Abbreviation in abstract (LFTs) without explanation -> I would generally try to avoid abbreviations in the abstract
- Part 1. Definition and epidemiology of NAFLD
o I would include more recent epidemiologic data regarding numbers and also expectations of NAFLD/NASH prevalence.
- Part 2. Pathogenesis of NAFLD
o Maybe also differentiate between modifiable ((lifestyle /diet/exercise, comorbidities, drugs, alcohol etc.) and non-modifiable (age, sex, race/ethnicity, family history, genetics etc.) risk factors in the text and also on figure 1 as there is a lot of space left.
- Part 3. Natural history
o Maybe include the fact that numbers of NASH and advanced liver fibrosis are still increasing and in consequence it is NASH is expected to be one of the predominant or even the major indication for liver transplantation, highlighting the need to have good screening procedures for patients at risk to stop further progression. Maybe add the following literature:
European Association for the Study of the Liver. Electronic address eee, Clinical Practice Guideline P, Chair et al. EASL Clinical Practice Guidelines on non-invasive tests for evaluation of liver disease severity and prognosis - 2021 update. J Hepatol 2021; 75: 659-689. doi:10.1016/j.jhep.2021.05.025
- Maybe add absolute numbers worldwide and absolute numbers in risk populations to the figure as I think absolute numbers might be more impressive for any reader.
- 6. Screening for NAFLD in primary care: current recommendations
o I would recommend to highlight the role of ultrasound together with anamnesis, biomarkers and scoring systems. From my personal perspective and clinical experience the use of ultrasound diagnostic is by far more important than the authors mentioned.
- I would recommend discussing/highlighting the following aspects:
o Screening of the general population is probably not useful regarding the high prevalence but should definitely focus patients at risk (T2DM, metabolic syndrome, obesity, arterial hypertension) as they have higher risk for progression to advanced liver fibrosis, which in turn represents the major risk for hepatic as well as extrahepatic complications, morbidity and mortality.
o Is it possible to provide another figure / chart for a screening algorithm based on the current review that might be useful for any clinicians. I personally recommend something like the algorithm from the german guidelines:
https://www.dgvs.de/wp-content/uploads/2022/10/LL-NAFLD_Leitline-englisch_zfg.pdf
Author Response
Reviewer 4.
In this literature review entitled “Screening for NAFLD – current knowledge and challenges” the authors provide a current update on NAFLD screening algorithms in the primary care setting and present as well as future challenges. The manuscript is well written and I have only minor comments or recommendations:
- Abbreviation in abstract (LFTs) without explanation -> I would generally try to avoid abbreviations in the abstract
We have addressed this comment.
- Part 1. Definition and epidemiology of NAFLD
o I would include more recent epidemiologic data regarding numbers and also expectations of NAFLD/NASH prevalence.
We have addressed this comment and included more numbers and projections in the Definition and epidemiology paragraph (page 1, line 35-40).
- Part 2. Pathogenesis of NAFLD
o Maybe also differentiate between modifiable ((lifestyle /diet/exercise, comorbidities, drugs, alcohol etc.) and non-modifiable (age, sex, race/ethnicity, family history, genetics etc.) risk factors in the text and also on figure 1 as there is a lot of space left.
We have addressed this comment and divided modifiable and non-modifiable factors in paragraph n.2 (page 2). The authors did not modify the Figure 1 as not all the risk factors mentioned in the figure can be clustered as either modifiable or not (i.e. lipotoxicity, hepatocellular injury, immunity)
- Part 3. Natural history
o Maybe include the fact that numbers of NASH and advanced liver fibrosis are still increasing and in consequence it is NASH is expected to be one of the predominant or even the major indication for liver transplantation, highlighting the need to have good screening procedures for patients at risk to stop further progression. Maybe add the following literature:
We have included numbers on liver transplantation in Definition and epidemiology (page 1, line 35-40). We have also highlighted the importance of early detection of liver disease in light of no medications being available at the moment (page 8, line 203-208).
European Association for the Study of the Liver. Electronic address eee, Clinical Practice Guideline P, Chair et al. EASL Clinical Practice Guidelines on non-invasive tests for evaluation of liver disease severity and prognosis - 2021 update. J Hepatol 2021; 75: 659-689. doi:10.1016/j.jhep.2021.05.025
Thank you for the suggestion. This paper was included among the references.
- Maybe add absolute numbers worldwide and absolute numbers in risk populations to the figure as I think absolute numbers might be more impressive for any reader.
Thank you for the suggestion. We have added more absolute numbers in the page 1, line 35-40.
- 6. Screening for NAFLD in primary care: current recommendations
o I would recommend to highlight the role of ultrasound together with anamnesis, biomarkers and scoring systems. From my personal perspective and clinical experience the use of ultrasound diagnostic is by far more important than the authors mentioned.
Dear reviewer, thank you for your comments. We followed guidelines indications which suggest that US has a limited role in screening as low sensitivity and specificity for screening NAFLD.
- I would recommend discussing/highlighting the following aspects:
o Screening of the general population is probably not useful regarding the high prevalence but should definitely focus patients at risk (T2DM, metabolic syndrome, obesity, arterial hypertension) as they have higher risk for progression to advanced liver fibrosis, which in turn represents the major risk for hepatic as well as extrahepatic complications, morbidity and mortality.
Thank you for the comments. The need for screening high-risk groups is highlighted in paragraph.
o Is it possible to provide another figure / chart for a screening algorithm based on the current review that might be useful for any clinicians. I personally recommend something like the algorithm from the german guidelines:
Thank you for the comments. We have included figure 3.
Reviewer 5 Report
Brief Summary
The article by Forlano et al offered a nice review of NAFLD, including the epidemiology, pathogenesis, diagnosis, staging, screening, etc. The article reads clear and well organized. The review discussed screening options and limitations and eventually calls for optimization of existing NAFLD screening methods. The article overall provides a nice reference for future researchers to advance the field.
General Concept Comments
1. It would significantly enhance the article to add a dedicated section at the end to discuss the current and promising future options of NAFLD treatment. This could include both lifestyle management (dietary, exercising, etc.) and pharmacological interventions (small molecules, RNAi, protonophore, etc.).
Specific Comments
None
Author Response
Dear reviewer, thank you for suggestions. We mentioned the crucial importance of early detection of NAFLD especially as pharmacological treatment is lacking (page 8,line 203-208). However, the authors feel that preparing a separate chapter on all the drugs in the pipeline goes beyond the scope of the review.